# Identification of Hub Genes and Potential Biomarkers for Childhood Asthma by Utilizing an Established Bioinformatic Analysis Approach

**DOI:** 10.3390/biomedicines10092311

**Published:** 2022-09-16

**Authors:** Ichtiarini Nurullita Santri, Lalu Muhammad Irham, Gina Noor Djalilah, Dyah Aryani Perwitasari, Yuniar Wardani, Yohane Vincent Abero Phiri, Wirawan Adikusuma

**Affiliations:** 1Faculty of Public Health, Universitas Ahmad Dahlan, Yogyakarta 55164, Indonesia; 2Faculty of Pharmacy, Universitas Ahmad Dahlan, Yogyakarta 55164, Indonesia; 3Medical Faculty Muhammadiyah Surabaya, Surabaya 60115, Indonesia; 4School of Public Health, College of Public Health, Taipei Medical University, Taipei 11031, Taiwan; 5Institute for Health Research and Communication (IHRC), Lilongwe P.O. Box 1958, Malawi; 6Departement of Pharmacy, University of Muhammadiyah Mataram, Mataram 83127, Indonesia

**Keywords:** bioinformatics, biomarkers, childhood asthma, genome-wide association study, hub genes

## Abstract

Childhood asthma represents a heterogeneous disease resulting from the interaction between genetic factors and environmental exposures. Currently, finding reliable biomarkers is necessary for the clinical management of childhood asthma. However, only a few biomarkers are being used in clinical practice in the pediatric population. In the long run, new biomarkers for asthma in children are required and would help direct therapy approaches. This study aims to identify potential childhood asthma biomarkers using a genetic-driven biomarkers approach. Herein, childhood asthma-associated Single Nucleotide Polymorphisms (SNPs) were utilized from the GWAS database to drive and facilitate the biomarker of childhood asthma. We uncovered 466 childhood asthma-associated loci by extending to proximal SNPs based on *r*^2^ > 0.8 in Asian populations and utilizing HaploReg version 4.1 to determine 393 childhood asthma risk genes. Next, the functional roles of these genes were subsequently investigated using Gene Ontology (GO) term enrichment analysis, a Kyoto Encyclopedia of Genes and Genomes (KEGG) pathway, and a protein–protein interaction (PPI) network. MCODE and CytoHubba are two Cytoscape plugins utilized to find biomarker genes from functional networks created using childhood asthma risk genes. Intriguingly, 10 hub genes (*IL6*, *IL4*, *IL2*, *IL13*, *PTPRC*, *IL5*, *IL33*, *TBX21*, *IL2RA*, and *STAT6*) were successfully identified and may have been identified to play a potential role in the pathogenesis of childhood asthma. Among 10 hub genes, we strongly suggest *IL6* and *IL4* as prospective childhood asthma biomarkers since both of these biomarkers achieved a high systemic score in Cytohubba’s MCC algorithm. In summary, this study offers a valuable genetic-driven biomarker approach to facilitate the potential biomarkers for asthma in children.

## 1. Introduction

Childhood asthma is the most frequent condition characterized by chronic airway inflammation [1]. In particular, asthma exacerbations are a significant cause of childhood morbidity and mortality. Asthma attacks frequently occur, sometimes leading to progressive lung damage or fatal in mild asthmatics. Exacerbation prevention becomes a top priority in treating all asthmatics [2]. The death rate of asthma in Children reached 0.7 per 100,000 people. This trend was predicted to increase and reach 400 million in 2025, consisting of adults and children. Asthma is the most prevalent chronic disease affecting children, and it is among the top 20 conditions in the world for disability-adjusted life years in children [3,4]. In addition, asthma prevalence has been rising over the past few decades, and differences in rates between nations strongly imply that environmental exposures play a significant role in asthma development. Asthma risk factors associated with the environment include tobacco smoke, farm animals and their products, domestic cats, respiratory viral infections, microbial exposures, dietary factors, breastfeeding, medication, occupational exposures, indoor and outdoor air pollution, and various allergens [5]. Environmental exposures begin in early childhood and play a crucial role as determinants of health. When these exposures occur at critical stages of development, they influence genetic responses and individual risk profiles that ultimately contribute to asthma development [6,7]. Genetic and environmental variables influence asthma’s pathophysiology and treatment effectiveness [8]. Developing an understanding of gene–environment interactions may aid in elucidating the cause of illness and classifying specific genes or exposures in the same pathway [9].

Over the past few decades, significant achievement has been made in understanding and treating asthma, including developing biomarkers and innovative targeted therapies [8]. However, the causes of and effective therapy for childhood severe asthma exacerbations only remain known at a low level [10]. In addition, severe asthma unresponsive to inhaled corticosteroids (ICS) therapy results in poor symptom control and increased exacerbations [11]. Since 2019, personalized management for children (6–11 years old) with asthma has already been mentioned in Global Initiative for Asthma (GINA), focusing on using Leukotriene receptor antagonist (LTRA) as the alternative to the low dose inhaled corticosteroid and Short acting β_2_ agonists [12]. The discovery of new asthma treatments can be discovered by a better understanding of the mechanism of different biomarkers and phenotypes [13]. Identifying biomarkers may provide new approaches to managing and treating childhood asthma [14].

A reliable genetic marker could identify and aid in treating childhood asthma more quickly. The discovery of new genetic targets can also provide new insights into the pathophysiology of childhood asthma [14]. Genetic variations that influence susceptibility to childhood asthma could lead to the development of novel therapeutics [15]. Several potential loci have been implicated in the genetics of childhood asthma, but no robust and specific markers for the disease have been identified [1]. Therefore, we systematically integrated several genomic databases to drive biomarkers for childhood asthma. This study used the Genome-Wide Association Study (GWAS) Catalog database to identify genetic markers associated with particular childhood asthma. Next, we perform functional enrichment and protein–protein interaction (PPI) network analyses to identify how these genes interact and cooperate to drive childhood asthma development. In addition, the Cytohubba plugin of Cytoscape provides 11 topological methods to identify some key genes. Maximal Clique Centrality (MCC) was selected in this study to predict essential proteins from the yeast PPI network more accurately among the eleven methods. Several articles have been published using the CytoHubba to identify potential biomarkers in different diseases [16,17,18,19]. In short, a combination of the data could enable the identification of novel childhood asthma as biomarkers and provide new insight into the underlying cellular mechanism of this debilitating condition.

## 2. Methods

### 2.1. Identified Childhood Asthma Risk Genes

We extracted the childhood asthma-associated SNPs by using National Human Genome Research Institute (NHGRI) (EMBL-EBI, Wellcome Genome Campus, Hinxton, Cambridgeshire, CB10 1SD, UK) GWAS catalog database (http://www.ebi.ac.uk/gwas) [20] accessed on 14 April 2022. The SNPs were chosen based on the disease/trait attribute of “Childhood asthma.” Next, HaploReg version 4.1 is used to expand the number of SNPs *r^2^* > 0.8 in Asian populations, indicative of a childhood asthma risk gene. Detailed information on the study workflow is depicted in Figure 1.

### 2.2. Gene Ontology Enrichment Analysis

Data analyzed for gene ontology enrichment analysis were obtained from the WEB-based Gene SeT Analysis Toolkit (WebGestalt) 2019 [21]. In general, gene ontology (GO) was divided into three categories as follows: biological process (BP), cellular component (CC), and molecular function (MF) [22]. The GO database contains annotations that explain the features of genes and gene products from various organisms and putative functions for enriched genes. BP is an ordered set of molecular functions that describe various biological processes. The MF is used to explain a gene’s or gene product’s function, while the CC describes genes’ subcellular structures, locations, and macromolecular complexes [16]. The significance threshold was set at a *q*-value (FDR) *<* 0.05.

### 2.3. KEGG Pathway Enrichment Analysis

The Kyoto Encyclopedia of Genes and Genomes (KEGG) pathway enrichment analysis was performed using the WebGestalt 2019 online tools regarding the KEGG database on candidate genes to enrich significantly altered pathways. The KEGG database links genomics information with higher-order functional information to systematically analyzed gene functions [23]. KEGG enrichment results are presented with *q*-values (FDR) < 0.05, indicating significance.

### 2.4. Discovering Biomarker Gene of Childhood Asthma

A protein–protein interaction (PPI) analysis was conducted based on candidate genes and proteins in the STRING database (https://string-db.org/), accessed on 14 April 2022. The STRING database is designed to provide a comprehensive listing of all known and predicted protein–protein interactions, including physical and functional associations [24]. STRING’s default settings were used. The PPI network was built and visualized using Cytoscape software version 3.7.2 (Bethesda, MD, USA) [25], accessed on 5 August 2022. Cytoscape is a graphical network visualization tool for complicated biological networks. We also utilized the Cytoscape plugin molecular complex detection (MCODE) to screen and identify important modules in the PPI network using the following scores and parameters: k score = 2, degree cutoff = 2, node score cutoff = 0.2, and maximum depth = 100 [26]. Next, using the CytoHubba plugin in Cytoscape, we calculated and analyzed the network structure of the PPI network to identify hub genes. CytoHubba features eleven topological analysis methods, including Degree (Deg), Edge Percolated Component (EPC), Maximum Neighborhood Component (MNC), Density of Maximum Neighborhood Component (DMNC), Maximal Clique Centrality (MCC), and six centralities based on shortest paths (Bottleneck, EcCentricity, Closeness, Radiality, Betweenness, and Stress). Among the eleven methods, we used the MCC algorithm to predict essential proteins more accurately from the yeast PPI network [27]. The top gene from the MCC algorithm was considered as a potential biomarker gene.

### 2.5. Statistical Analysis

Our study used RStudio 4.2.1 (RStudio, 250 Northern Ave, Boston, MA 02210, USA) for all analytic workflows. Over-representation analysis (ORA), including gene ontology (GO) and KEGG pathway enrichment analysis, were performed using the WebGestalt 2019 R package [28]. GO and KEGG were visualized using R with the ggplot2 package [29]. In addition, STRING and Cytoscape were used to construct and visualize the PPI [25].

## 3. Results

### 3.1. Childhood Asthma Risk Genes Identification

The GWAS catalog yielded a total of 466 childhood asthma-associated SNPs with criterion *p*-value < 10^−5^ (Appendix A). Following that, we utilized HaploReg version 4.1 (Massachusetts Institute of Technology, 77 Massachusetts Avenue, Cambridge, MA, USA) to determine the proxy SNPs of childhood asthma-associated SNPs. Interestingly, our study emphasized that 393 childhood asthma risk genes overlap with the childhood asthma-associated SNPs, and it is based on the characteristic of *r*^2^ > 0.8 used in Asian populations (Appendix A).

### 3.2. Gene Ontology Enrichment Analysis of Childhood Asthma Risk Genes

In order to analyze the biological features of the discovered genes and proteins, GO enrichment analysis was done using the WebGestalt 2019 online tools. The GO enrichment analysis includes a biological process (BP), cellular component (CC), and molecular function (MF). For each GO enrichment analysis, the significance level was set at a *q*-value (FDR) < 0.05. The analysis of BP showed that a total of 471 functions were significantly enriched. The top-ranked results indicated that BP is primarily associated with “regulation of immune system process”, “cellular response to cytokine stimulus”, “positive regulation of immune system process”, “cytokine-mediated signaling pathway”, and “response to cytokine production”. The CC analysis included 39 functions significantly enriched, such as “MHC class II protein complex”, “integral component of luminal side of endoplasmic reticulum membrane”, luminal side of endoplasmic reticulum membrane”, “MHC protein complex”, and “side of membrane”. The MF analysis revealed that 30 functions were significantly enriched, including “peptide antigen binding”, “MHC class II receptor activity”, “signaling receptor binding”, “cytokine activity”, and “interleukin-1 receptor activity”. Figure 2 depicts the top ten GO enrichment analyses (BP, MF, CC). The complete results of the GO enrichment analysis are shown in Appendix A.

### 3.3. KEGG Pathway Analysis of Childhood Asthma Risk Genes

KEGG enrichment pathway analysis was performed using WebGestalt 2019 to establish the potential involvement of pathways related to identified gene candidates. The KEGG analysis revealed 34 significantly changed pathways (*q* < 0.05) (Appendix A). Most of the highly ranked pathways were involved in categories such as “Th17 cell differentiation”, “inflammatory bowel disease (IBD)”, “Th1 and Th2 cell differentiation”, “allograft rejection”, and “asthma”. The KEGG pathway enrichment analysis results for the top 10 ranks are illustrated in Figure 3.

### 3.4. Identification of Potential Biomarkers of Childhood Asthma

Protein–protein interaction (PPI) network of 393 childhood asthma risk genes was constructed by using the STRING database (Figure 4). In addition, biomarker genes were isolated from the PPI networks using Cytoscape plugins such as MCODE and CytoHubba. The Cytoscape plugin MCODE was used to identify gene clusters (probable biomarkers) from the PPI networks. The PPI network was subclustered into eight subclusters using the MCODE method (Figure 4). Appendix A contains a comprehensive list of MCODE clusters, including their score, number of nodes, and edges. Next, we utilized CytoHubba to choose hub genes for the PPI network. The hub genes are the nodes of the network with high connectivity. Cytohubba’s MCC algorithm was used to rank all nodes. We highlighted that 10 biomarkers (*IL6*, *IL4*, *IL2*, *IL13*, *PTPRC*, *IL5*, *IL33*, *TBX21*, *IL2RA*, and *STAT6*) were identified as the highest 10 ranked hub genes (Table 1). Among these biomarkers, we emphasized that *IL6* and *IL4* were the top two of the ten hub genes and were determined to represent the key genes and considered potential biomarkers of childhood asthma.

## 4. Discussion

GWAS or through gene-environment interactions can be used to uncover the genetic variance associated with asthma and identify possible targets for therapeutics [30]. This study focused on identifying potential biomarkers for childhood asthma based on candidate genes from the GWAS-identified loci. A GWAS is a comprehensive analysis of all genetic variations in the form of SNPs that can be associated with certain traits. GWAS can help detect genetic biomarkers associated with common, complex diseases or phenotypes [20,31,32,33,34,35]. An integrated bioinformatics analysis was performed in this study based on candidate genes identified from GWAS loci, GO enrichment analyses, KEGG pathway enrichment analyses, and a constructed PPI network. In this way, our findings provide valuable clues for efforts to investigate the potential biomarker for diagnosis and treatment of childhood asthma. The discovery of new biomarkers aid in the categorization of patients, therapeutic response, and clinical outcome prediction [36]. Most biomarkers for asthma are confined to the Th2 phenotype, and no useful biomarkers for severe asthma have been confirmed in both children and adults [37]. Our work identified ten potential biomarkers in childhood asthma, including *IL6*, *IL4*, *IL2*, *IL13*, *PTPRC*, *IL5*, *IL33*, *TBX21*, *IL2RA*, and *STAT6*. Among these, *IL6* and *IL4* were screened out as highly potential biomarkers in childhood asthma since the gene also acquired a high systemic score in Cytohubba’s MCC algorithm.

Interleukin 6 (*IL6*) was recently identified as a promising biomarker for adult asthma in peripheral blood [38]. Nevertheless, *IL6* has not been studied as a biomarker in childhood asthma [39]. *IL6* promotes effector T cells, suggesting that it may have a functional role in asthma. The cytokine *IL6* is traditionally thought of as an inflammatory marker, along with TNFα and IL-1β, instead of a regulatory cytokine [40]. *IL6* levels are associated with systemic inflammation, metabolic dysfunction, and greater asthma severity among lean and obese adults [38]. Elevated *IL6* levels were associated with reduced lung function and heightened asthma exacerbation risk. In addition, epithelial *IL6* trans-signaling has been identified as a potential mechanism linked to asthma phenotypes characterized by increased airway inflammation [41]. Interestingly, *IL6* increased asthma exacerbation risk in children but did not affect lung function or other severity indicators as in adults. It is possible that children with frequent exacerbations and high *IL6* levels may grow up to be adults with severe asthma [39]. Evidence from a longitudinal study indicates that children with higher plasma *IL6* levels are related to obesity, metabolic syndrome, and greater asthma severity, with a risk for asthma exacerbation and decreased lung function [42].

Allergies and acute asthma exacerbations are generally induced by *IL4* [43]. *IL4* is an important cytokine involved in asthma development [44], and it has been reported that airway hyperresponsiveness, eosinophil infiltration, and inflammation are all symptoms of bronchial asthma that *IL4* likely mediates [45]. *IL4* is a potent activator of inflammation and is involved in developing fibrosis during Th2 inflammation [46]. In asthma patients, Th2 is hyperactive, causing a rise in *IL4* and immunoglobulin E (IgE), which stimulates the growth and activation of eosinophilic granulocytes, which then secretes a variety of inflammatory mediators, leading to bronchial chronic inflammation and asthma [47]. In addition, in asthmatic children, levels of *IL4* were significantly higher among atopic asthmatics than nonatopic asthmatics [48,49]. *IL4* and exhaled nitric oxide, combined with 8-isoprostane and interferon-gamma (*IFN-**ɤ*), were effective markers for asthma control. Children with asthma also had a higher *IL4*/*IFN-**ɤ* ratio, consistent with a predominant Th2 inflammatory state [49,50]. *IL4* is an anti-inflammatory protein that can modulate inflammation, thus preventing asthma and vice versa [44]. In particular, we would like to emphasize that our method of genetic-driven biomarkers ultimately provided a candidate list of the biomarker in childhood asthma to be used in clinical outcome predictions. Nevertheless, further research is required to verify this finding.

Our study has some limitations which need careful consideration. The results presented here are based on current information obtained from GWAS Catalog. In the future, new information may influence the results. Since we relied on existing data for integrative bioinformatic analysis, further information may affect the presented results. Additionally, our analyses have primarily been exploratory; therefore, further confirmation in the functional study is still required.

## 5. Conclusions

In summary, our findings point to possible biomarkers for asthma in children. These bioinformatic studies reveal significant hub genes such as *IL6*, *IL4*, *IL2*, *IL13*, *PTPRC*, *IL5*, *IL33*, *TBX21*, *IL2RA*, and *STAT6*, which may play a pivotal role in asthma development in children. We highlighted that *IL6* and *IL4* obtained a high systemic score in Cytohubba’s MCC algorithm. We therefore highly proposed both of these biomarkers as potential childhood asthma biomarkers. However, further study is needed to understand these genes’ regulatory functions better so that their relevance as clinical biomarkers and therapeutic targets may be established.

## Figures and Tables

**Figure 1 biomedicines-10-02311-f001:**
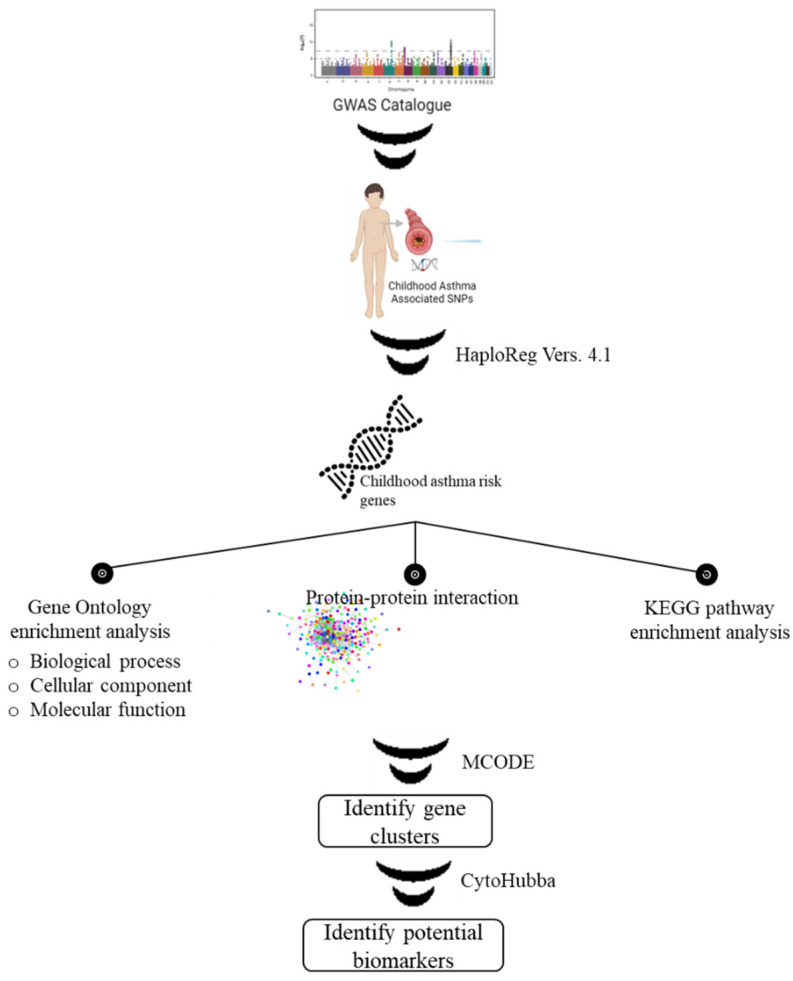
Study workflow of bioinformatics analysis. We integrated several genomic databases (GWAS and HaploReg) and several functional enrichment-based approaches (GO, PPI and KEGG) to identify potential biomarkers for childhood asthma.

**Figure 2 biomedicines-10-02311-f002:**
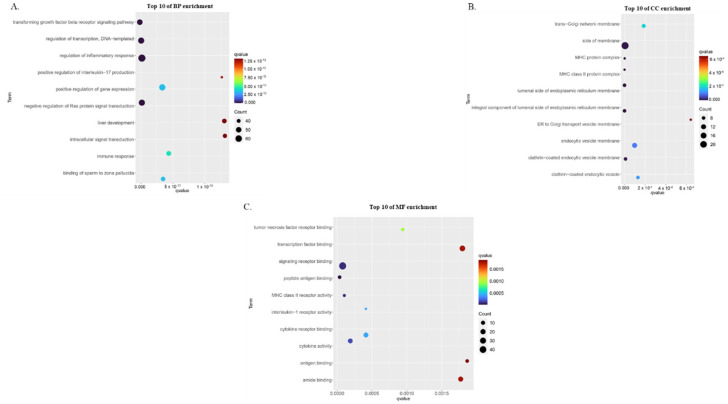
Gene ontology enrichment analysis of childhood asthma risk genes by WebGestalt 2019. (**A**) top 10 of biological process (BP) enrichment analysis; (**B**) top 10 of cellular component (CC) enrichment analysis; (**C**) top 10 of molecular function (MF) enrichment analysis.

**Figure 3 biomedicines-10-02311-f003:**
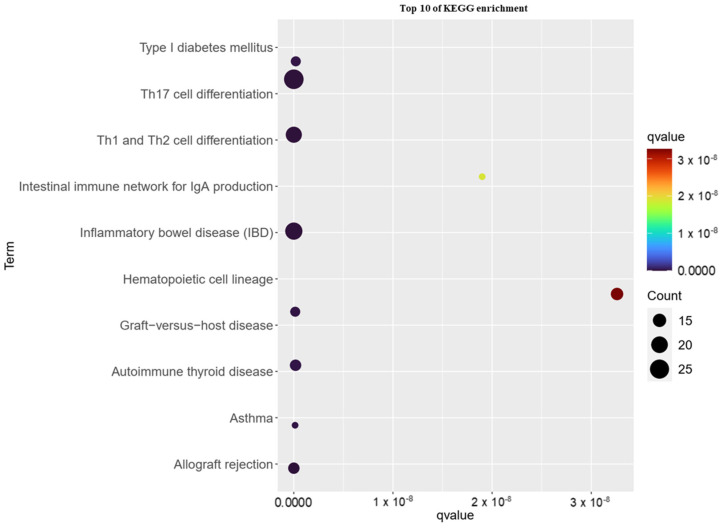
KEGG pathway enrichment analysis of childhood asthma risk genes by WebGestalt 2019.

**Figure 4 biomedicines-10-02311-f004:**
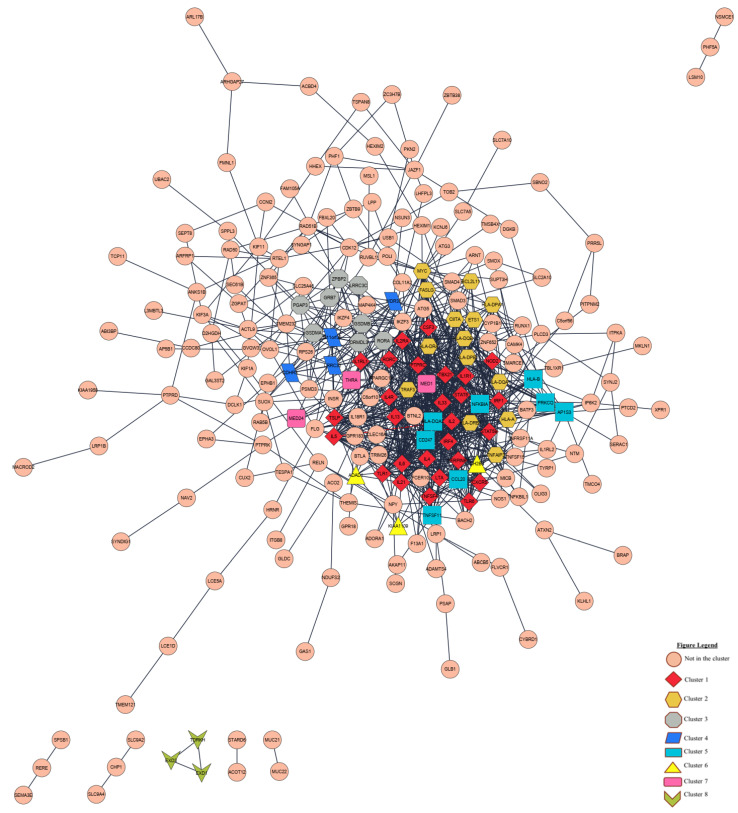
Protein–protein interaction (PPI) network of childhood asthma risk genes was constructed by STRING and Cytoscape. The PPI network had 237 nodes and 991 edges. Additionally, eight modules were detected by MCODE in Cytoscape. Cluster 1 (score = 19.615), cluster 2 (score = 5.538), cluster 3 (score = 5.143), cluster 4 (score = 3.333), cluster 5 (score = 3.143), cluster 6 (score = 3), cluster 7 (score = 3), and cluster 8 (score = 3).

**Table 1 biomedicines-10-02311-t001:** The top 10 hub genes identified by using CytoHubba.

Rank	Gene ID	Gene Name	Score
1	*IL6*	Interleukin 6	22,370,717,281
2	*IL4*	Interleukin 4	22,370,701,130
3	*IL2*	Interleukin 2	22,370,622,650
4	*IL13*	Interleukin 13	22,369,670,594
5	*PTPRC*	Protein Tyrosine Phosphatase Receptor Type C	22,363,673,434
6	*IL5*	Interleukin 5	22,340,430,000
7	*IL33*	Interleukin 33	21,784,097,718
8	*TBX21*	T-Box Transcription Factor 21	21,767,214,138
9	*IL2RA*	Interleukin 2 Receptor Subunit Alpha	21,701,358,536
10	*STAT6*	Signal Transducer and Activator of Transcription 6	21,233,178,024

The higher score, the higher rank of the biomarker. The score of genes was identified from CytoHubba.

## Data Availability

The data presented in this study are available in Appendix A here.

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
