# Peer review of "Identification of Hub Genes and Potential Biomarkers for Childhood Asthma by Utilizing an Established Bioinformatic Analysis Approach"

_biomedicines, 2022, doi:10.3390/biomedicines10092311_

Round 1

Reviewer 1 Report

In this study, the authors focused on identifying potential biomarkers for childhood asthma based on candidate genes from the GWAS-identified loci. The authors identified IL6 and IL4 as biomarkers of potential childhood asthma. The results of this study are meaningful, but some questions remain.

1. Why was the CytoHubba tool used for biomarker discovery? What were the advantages of this method? Were there other more appropriate methods? These questions need to be explained clearly in the INTRODUCTION.

2. How can IL6 and IL4 be used as markers since they are widely distributed in the human body?

3. Biomarkers must correspond to their modification, e.g., gene mutations, overexpression of genes, amount of metabolite, etc. So what are the modifications corresponding to IL6 and IL4 as biomarkers? Does the presence of IL6 or IL4 indicate childhood asthma? Or do high levels of IL6 or IL4 indicate childhood asthma?  

Reviewer 2 Report

The manuscript “Identification of hub genes and potential biomarkers for childhood asthma by utilizing an established bioinformatic analysis approach” describes a search for Single Nucleotide Polymorphisms (SNPs) associated with childhood asthma in Asian populations in a GWAS database. The reason for the search was the potential to find reliable genetic markers to identify asthma and start therapy earlier. Another reason to search for biomarkers was to find targets for new treatment options. The identified biomarkers were IL6 and IL4.

The methods are carefully performed and well described and the conclusions based on the results. However, there are some concerns with the study:

Major concerns:

The results are not surprising or new. We already knew this. How well do these biomarkers identify asthmatics to help start therapy earlier? With what rates do these biomarkers give false positives/negatives? The usefulness of the results needs to be further discussed.

Can IL6 and IL4 be used as therapy targets in children? This question needs to be discussed.

The goal is the Asian population, but can the same biomarkers be used in other populations? This needs to be addressed.

Minor concerns:

The language needs to be improved. Some examples:

Abstract: Hence, new biomarkers for diagnosing and predicting therapy responses for childhood asthma are emergence needed.

What do you mean by this sentence? Are biomarkers urgently needed, or what is the message here?

Line 53: However, childhoods with severe asthma exacerbation are still poorly understood and treated significantly.

What do you mean by this statement? Please rephrase to clarify the point.

Next sentence: In addition, high-dose inhaled corticosteroid therapy results in severe asthma patients who are unsatisfactory, with uncontrolled symptoms and frequent exacerbations.

The patients are not unsatisfactory, I hope. Please rephrase.

The introduction lacks focus and the language needs improvement.

Round 2

Author Response

We sincerely thank you for the reviewer's comments and suggestions.

Reviewer 2 Report

The changes made to the manuscript have improved the understanding, but there are still errors in the language. Please do not use contracted form as this is a scientific text. There are also some sentences in need of explanation:

Line 44: "amongst the top 20 countries in the world." What do you mean by top 20 countries? Top in regard to what?

Line 63: What is GINA?

Line 81 and 220: What is MCC/MCC algorithm?

Author Response

Please see the file attached. Thank you
